# Low seroprevalence of Ebola virus in health care providers in an endemic region (Tshuapa province) of the Democratic Republic of the Congo

**Trésor Zola Matuvanga**[1,2,3]*, **Joachim Mariën**[4], **Ynke Larivière**[2,3], **Bernard Isekah Osang'ir**[2,3], **Solange Milolo**[1], **Rachel Meta**[1], **Emmanuel Esanga**[5], **Vivi Maketa**[1], **Junior Matangila**[1], **Patrick Mitashi**[1], **Steve Ahuka Mundeke**[6], **Hypolite Muhindo-Mavoko**[1], **Jean-Jacques Muyembe Tamfum**[6], **Pierre Van Damme**[2], **Jean-Pierre Van Geertruyden**[3]

**1** Tropical Medicine Department, University of Kinshasa, Kinshasa, Kinshasa, Democratic Republic of the Congo, **2** Vaccine and Infectious Disease Institute, Centre for the Evaluation of Vaccination, University of Antwerp, Wilrijk, Antwerp, Belgium, **3** Department of Family Medicine and Population Health, Global Health Institute, University of Antwerp, Wilrijk, Antwerp, Belgium, **4** Department of Biology, Evolutionary Ecology Group, University of Antwerp, Wilrijk, Antwerp, Belgium, **5** Division Provinciale de la Santé de la Tshuapa, Ministry of Health Hygiene and Prevention, Boende, Tshuapa, Democratic Republic of the Congo, **6** Department of Virology, Institut National de Recherches Biomedicales, Kinshasa, Kinshasa, Democratic Republic of the Congo

* zola.matuvanga@unikin.ac.cd, zolanga@yahoo.fr

**Data Availability Statement:** All relevant data are within the Supporting information files (S1, S2 and S3).

## Abstract

### Introduction

A serosurvey among health care providers (HCPs) and frontliners of an area previously affected by Ebola virus disease (EVD) in the Democratic Republic of the Congo (DRC) was conducted to assess the seroreactivity to Ebola virus antigens.

### Methods

Serum samples were collected in a cohort of HCPs and frontliners (n = 698) participants in the EBL2007 vaccine trial (December 2019 to October 2022). Specimens seroreactive for EBOV were confirmed using either the Filovirus Animal Nonclinical Group (FANG) ELISA or a Luminex multiplex assay.

### Results

The seroreactivity to at least two EBOV-Mayinga (m) antigens was found in 10 (1.4%: 95% CI, 0.7–2.6) samples for GP-EBOV-m + VP40-EBOV-m, and 2 (0.3%: 95% CI, 0.0–1.0) samples for VP40-EBOV-m + NP-EBOV-m using the Luminex assay. Seroreactivity to GP-EBOV-Kikwit (k) was observed in 59 (8.5%: 95%CI, 6.5–10.9) samples using FANG ELISA.

### Conclusion

In contrast to previous serosurveys, a low seroprevalence was found in the HCP and front-line population participating in the EBL2007 Ebola vaccine trial in Boende, DRC. This

**Funding:** • Grant Recipient: Pierre Van Damme and Jean-Pierre Van geertruyden • The full name of each funder: Innovative Medicines Initiative 2, European Union's Horizon 2020 research and innovation pro-gramme, European Federation of Pharmaceutical Industries and Associations (EFPIA) and the Coalition for Epidemic Preparedness Innovations (CEPI) • URL of each funder website: https://www.imi.europa.eu/about-imi https://ec.europa.eu/research-and-innovation/en/horizon-magazine/mission-transform-our-cities?gclid=Cj0KCQjwmZejBhC_

underscores the high need for standardized antibody assays and cutoffs in EBOV serosurveys to avoid the broad range of reported EBOV seroprevalence rates in EBOV endemic areas.

## Introduction

Ebola virus disease (EVD) was first observed during two simultaneous epidemics in 1976 in South Sudan and the Democratic Republic of the Congo (DRC) [1,2]. Since then, there were fifteen epidemics throughout the DRC, including one in the Boende area, province of Tshuapa in 2014 [3,4]. The frequency of EVD epidemics in the DRC increased tremendously over the past five years with seven epidemics occurring between 2017 and 2022. Mathematical models predict at least one epidemic each year [5]. An improved surveillance system and better diagnostic tools can partly explain the increasing trend. Further, villagers are more likely to come into contact with the natural reservoir of Ebolavirus Zaire (EBOV), as pristine habitats in the Congo Basin are transformed into farmland and cut at an unprecedented rate to provide wood for industries [6]. Human encroachment into these new habitats results in increased bushmeat hunting and a higher level of exposure to the virus, which is most likely spilled over from bats or monkeys [7]. Furthermore, flare-ups of EVD epidemics might also result from chronically infected patients, which was noted recently in Guinea where a survivor passed the virus on to his partner via semen more than 500 days after contracting EVD [5,8,9].

While spillover from animals to humans is considered to be rare [10], epidemics are primarily the result of direct person-to-person transmission via body fluids or indirect transmission via contaminated materials [11]. Due to occupational exposure, healthcare providers (HCPs) are more at risk during an outbreak than others in the general community and become a potential source of transmission themselves [12]. For example, during the seventh EVD outbreak in the DRC, which occurred in Boende Health District (2014), three HCPs were identified as potential super-spreaders of community-level disease transmission [13]. Similarly, health facilities may facilitate transmission to the community as infected patients, visitors, and the general public come together there [14,15]. For example, the EBOV epidemic in 1995 was mainly driven by nosocomial transmission at Kikwit General Hospital of DRC [16].

While epidemics are typically monitored through PCR-confirmed active cases, serosurveillance data represents the accumulative number of infections and may detect several undiagnosed cases. Indeed, EBOV infections may remain asymptomatic or paucisymptomatic after exposure to the pathogen [17]. This has been observed in recent studies where EBOV antigen seroreactivity is increasingly reported [8,18,19]. In unaffected areas, seroreactivity to GP-EBOV was reported in urban areas of Cameroon (1.3%), and DRC in Kinshasa (2%) and Kasaï Oriental (3.5%) [8,18,20,21]. In a resident pygmy population including traditional hunters in Watsa locality (Haut-Uele province, DRC) a seroprevalence of 18.7% was reported [8]. A study including HCP and frontliners, regardless of their self-reported history of EVD, found 3.4% of EBOV antigens seroreactivity in Kabondo—Dianda (southeastern DRC and forest-savannah area) [22]. A serosurvey conducted at the end of the 2014–2016 epidemic in Sierra Leone showed a seroreactivity of 8% among apparently healthy participants volunteering for an Ebola vaccine trial, with no self-reported history of EVD [23]. Serosurveys in the DRC obtained highly variable seroprevalence estimates depending on the region and the target group. While the EBOV seroprevalence in Boende after the previous epidemic of 2014 was high (28.1%) among healthy HCP never reporting an infection [24], the seroprevalence estimate was much lower in another study conducted in the same area (7%) [22]. A serosurvey

**Table 1. EBOV seroprevalence estimates using different assays in DRC.**

| Area of DRC | Year | *EBOV Seroprevalence (%) | Assay | CI | Population | Sample size (N) | Studies |
|---|---|---|---|---|---|---|---|
| Kikwit | 1995 | 2.2 | ELISA | 0.3–4.0 | Forest and City Workers | 414 | Busico et al. *Journal of Infectious Diseases* 1999, 79: S102-S107. |
| Watsa | 2002 | 18.7 | ELISA | 14.4–23.5 | General popualtion (pygmy) | 300 | Mulangu et al. *BMC infectious diseases* 2016, 16.1: 1–6 |
| Sankuru | 2007 | 11.0 | ELISA | 9.9–12.7 | General population | 3415 | Mulangu et al. *The Journal of Infectious Diseases* 2018, 217.4: 529–537 |
| Kinshasa | 2011–2012 | 2.0 | Luciferase immunoprecipitation system + neutralization | 0.7–5.1 | Blood donors | 752 | Imke et al., *Emerging Infectious Diseases.* 25 (5) 2019 |
| Boende | 2015 | 22.5 | ELISA | 19.2–25.9 | Healthcare workers | 611 | Doshi et al. *The Journal of Infectious Diseases* (2020). |
| Boende | 2015 | 28.1 | ELISA, Luciferase immunoprecipitation system + neutralization | 24.4–31.4 | Healthcare workers | 565 | Hoff et al. *The Journal of infectious diseases*, 2019, 219.4: 517–525 |
| Boende | 2015–2017 | 7.0 | ELISA | 5.0–8.8 | General population | 687 | Bratcher et al. *PLoS Neglected Tropical Diseases*, 2021, 15.8: e0009566. |
| Beni, Butembo, Katwa, and Mabalako | 2018–2020 | 2.3 | Luminex assay | 1.1–4.0 | Suspected cases of the tenth DRC epidemic of Ebola | 600 | Nkuba-Ndaye et al. *J Infect Dis.* 2022;226(2):352–356 |

*Seroprevalence based on the GP-EBOV antigen seroreactivity.

conducted on blood samples collected from clinically suspected EVD cases that were sent home after testing negative in two consecutive EBOV RT-PCR during the tenth EBOV outbreak in DRC Ituri, Nord Kivu and Sud Kivu provinces, 2018–2020), reported an EBOV antigen seroreactivity of 2.3% [25] (Table 1).

However, despite many studies assessing the GP-EBOV antigen seroreactivity in different populations and different locations/countries, the interpretation of this seroprevalence data is challenging given the variation of the assays employed and diversity of cutoff algorithms used [8,12,18,26–28]. Seroreactivity to a single EBOV antigen may not be sufficient to demonstrate prior exposure to EBOV, especially in asymptomatically infected persons [29,30]. Despite the broad range of EBOV seroprevalence rates in the EBOV endemic areas, previous serological surveys may have overestimated seroprevalence rates due to cross-reactivity against other infectious diseases (i.e. low specificity) [10,28]. The use of more specific assays to determine the seroreactivity based on at least two antigens may therefore provide a better understanding of the baseline seroprevalence before a vaccine immunogenicity assessment [31,32].

The study presented here, combines (1) the seroresults of baseline blood samples collected among HCP and frontliners participating in the EBL2007 vaccine trial which evaluates the safety and immunogenicity of the two-dose Ad26.ZEBOV, MVA-BN-Filo Ebola virus vaccine regimen (ClinicalTrials.gov identifier: NCT04186000) with (2) the results from an ecological survey to determine information related to the current and past residence and work locations of a cohort of HCP included in the EBL2007 vaccine trial [33]. On the baseline blood samples collected, pre-existing antibodies against EBOV among the participants were assessed using both FANG ELISA and Luminex assay. While the first assay only targets IgG antibodies against the glycoprotein (GP) of EBOV, the second assay also targets the nucleocapsid (NP)

and the viral matrix protein 40 (VP40) which increases its specificity to 99% [34]. This manuscript reports the baseline seroprevalence of Ebolavirus Zaire (EBOV) among HCP and frontliners participants in the EBL2007 trial conducted in the health district of Boende in DRC.

## Materials and methods

### Origin of samples

Baseline serum samples were collected before vaccination in an open-label, monocentric, phase 2, randomized trial to evaluate the immunogenicity and safety of Ad26.ZEBOV and MVA-BN-Filo in healthy HCP and frontliners in Boende Health District of DRC (EBL2007 trial, ClinicalTrials.Gov: NCT04186000) [33]. The trial site was located in the Boende General Hospital of Tshuapa province at approximately 750 km north-west of the capital city of Kinshasa in DRC. Blood samples were collected from healthy participants with no reported history of EVD or previous EBOV vaccination. During the first visit of the EBL2007 trial serum samples were collected for baseline determination of IgG GP-EBOV by the means of FANG ELISA and Luminex assay. At one year after inclusion of participants in the EBL2007 vaccine trial, a survey nested within the EBL2007 vaccine trial collected information related to where HCPs and frontliners lived and worked in the past, and their previous contacts with EVD cases.

### Operational definition

The HCP term in the EBL2007 vaccine trial included medical doctors, nurses, midwives, laboratory staff, pharmacy staff, hygienists, health facility cleaners, and nursing assistants working in a hospital, Health Center, Health Post, or Health District office. Frontliners encompassed community health workers, first aiders, and those working in the Health District office and or the Provincial Division of Health. Direct contact was defined as any HCP and frontliners who may have had direct interaction with patients infected with EVD at a hospital or treatment center during an outbreak. Indirect contact was considered the work of frontliners and other HCPs whose jobs did not bring them into direct interaction with sick patients but could bring them in contact with contaminated material.

### Serological testing

The study was performed according to the good clinical laboratory practice guidelines of the Division of Acquired Immunodeficiency Syndrome and WHO [35,36] to ensure high quality, reliable, and reproducible data at Q Squared Solutions (San Juan Capistrano, CA, US) Vaccine Testing Laboratory for FANG ELISA and Institut National de Recherche Biomédicale (INRB) in DRC for the Luminex Assay.

Considering only the seroreactivity to GP EBOV antigen, a higher specificity (95.4%: IC95% 89.6–98.0) and similar sensitivity (96.8%: IC95 91.3–98.9) to that of commercial ELISA assays was reported in a study comparing Luminex to the commercial ELISA kits more commonly used in previous serological surveys [32]. Using the FANG ELISA was shown to be greater accurate and precise than a commercial alternative for assessing immune response after Ebola vaccination [37].

### LUMINEX Assay technology

The serology testing was performed with Luminex Magpix® technology (Luminex Corp., Austin, TX) as per the previously published protocol [17,32]. Four recombinant commercially available EBOV antigens were coated onto magnetic beads: two glycoproteins, GP-EBOV-kis (Kissidougou/Makona 2014 strain) and GP-EBOV-m (Mayinga 1976 strain); 1 nucleoprotein,

NP-EBOV-m (Mayinga 1976 strain); and 1 40-kDa viral protein (VP40-EBOV-m, Mayinga 1976 strain). The bead-coupled antigens were mixed with the patient sample (1:1000 sample to dilution buffer), and the signal from the response for anti-EBOV immunoglobulin G (anti-IgG) was read and stored on Bio-Plex 200 hardware (Bio-Rad, Marnes-la-Coquette, France). All results were reported as the median fluorescence intensity (MFI). Based on the serological responses, a participant was deemed to bear the pre-existing antibodies against EBOV antigens when the sample was reactive above the cutoff for at least two different EBOV antigens.

## FANG ELISA

The methods used to perform the FANG ELISA have been described in previous studies [37]. Before the addition of test samples, 96-well microplates were coated with 100 μL of recombinant GP-EBOV-Kikwit (k) and incubated at 4˚C in the absence of light. In addition to this, a standard obtained from one or more serially diluted vaccinated donors had been added. Incubation was performed by adding horseradish peroxidase conjugate from goat anti-human IgG to each well. The substrate 3, 3', 5, 5'-tetramethylbenzidine was then incorporated into each well. The addition of sulfuric acid solution stopped the enzymatic reaction. The color change was then observed with a plate reader. The plate reader was used to report the quality controls as well as the concentrations of the added samples. The concentrations of these samples were based on the standard curve calculated using a 4-parameter logistic curve (4PL) and are expressed as ELISA units/ml (EU/ml). Final titers were determined based on a cutoff optical density (OD) value and were reported as the reciprocal of the highest dilution with a positive OD value.

## Sample size and statistical analysis

The number of participants eligible for the EBL2007 trial with available aliquots (n = 698) at the inclusion visit predetermined the number of enrolled subjects in the serosurvey. Subjects reacting to EBOV antigens (GP, NP, and VP40) were summarized using proportions with 95% confidence interval. Demographic and ecological data were compiled and summarized using descriptive statistics for all participants enrolled in the EBL2007 vaccine trial using SPSS 28.0 IBM SPSS Statistics for Windows, version 28.0 and R 4.2.1 Statistical Software.

Both the FANG ELISA and Luminex assay do not have an established cutoff to distinguish individuals with seroreactivity to an EBOV antigen. In the absence of a represented control panel to estimate a cutoff, we calculated cutoff values by change point analysis [38] using R [39]. In the supporting information, we also provide seroprevalence estimates based on cutoff values obtained from literature (S4 Table).

To further investigate if the signal of the antibody assay represents true past exposure to EBOV, we tested if participants from the EBOV risk groups (based on age, sex, direct or indirect contact with patients in general, working in a hospital or elsewhere, previous contact with Ebola patients or experienced an outbreak at a location where you lived) were significantly more likely to be antibody positive. We used a generalized linear model with binomial link function. For each individual antigen, the participant's seropositivity status was included as response variable and the participant characteristics as explanatory variables. Only combinations of the Luminex GP-EBOV-m+VP40-EBOV were considered, as the sample size of the positive group was too small for all other combinations. P-values were considered significant below a value of 0.05.

## Ethics statements

Ethics Committee of the University Hospital of Antwerp/University of Antwerp (approval reference n˚19/14/177) and the National Ethics Committee of the DRC Ministry of Health approved

the study protocol of EBL2007 (approval reference n˚121/CNES/BN/PMMF/2019). The National Ethics Committee of the DRC Ministry Health under approval reference n ˚ 212/CNES/BN/ PMMF/2020 approved the ecological survey nested in the EBL2007 Vaccine trial. For both the EBL2007 trial and the ecological survey participants provided written informed consent.

# Results

## Participants characteristics

A total of 720 HCPs and frontliners were screened for inclusion in the EBL2007 trial, of which 699 (96.9%) agreed to participate in the baseline seroprevalence study. However, one participant withdrew consent prior to blood collection. Thus, blood samples were available for 698 (99.9%) participants with a mean age of 45 years (standard deviation = 12.0) and 534 (76.5%) were male (Table 2). The FANG ELISA results for five samples were indeterminate. Nearly two-thirds of the HCPs and frontliners [492 (70.5%)] worked in a health facility in the Boende Health District and 410 (59.0%) were HCPs working in direct contact with patients. Forty-three (6.2%) of them reported a direct contact with patients during a previous Ebola outbreak in Boende or elsewhere. From a minority (3.5%) we are not sure if they ever had contact with infectious patients during an Ebola outbreak.

## Seroreativity to EBOV proteins using FANG ELISA and or Luminex

When considering antibody responses against EBOV antigens individually, we found that 8.5% (60/698; 95%CI 6.5–10.7) of samples tested positive on the Luminex for GP-EBOV-m,

**Table 2. Participants characteristics.**

| Characteristic | N = 698 | % | Mean (SD) | Min | Max |
|---|---|---|---|---|---|
| **Age** (year) | | | 45.0 (12.0) | 19 | 75 |
| **Sex** | | | | | |
| Female | 164 | 23.5 | | | |
| Male | 534 | 76.5 | | | |
| **Profession** | | | | | |
| Community Health Worker | 236 | 33.8 | | | |
| Nurse | 181 | 25.9 | | | |
| First Aid Worker | 177 | 25.4 | | | |
| Hygienist | 37 | 5.3 | | | |
| Midwife | 30 | 4.3 | | | |
| Medical Doctor | 13 | 1.9 | | | |
| Health Facility Cleaner | 10 | 1.4 | | | |
| Care Giver | 7 | 1.0 | | | |
| Other | 3 | 0.4 | | | |
| Laboratory Technician | 2 | 0.3 | | | |
| Pharmacist Assistant | 2 | 0.3 | | | |
| **Place of work in Boende** | | | | | |
| Health Facility (Hôpital, Centre de Santé, Poste de Santé) | 492 | 70.5 | | | |
| Health District Office (Bureau central Zone de Santé) | 8 | 1.1 | | | |
| Croix-Rouge Boende | 177 | 25.4 | | | |
| Inspection Provinciale de la Santé | 1 | 0.1 | | | |
| Aire de Santé | 10 | 1.4 | | | |
| Division Provinciale de la Santé Tshuapa | 9 | 1.3 | | | |
| Programme Elargi de Vaccination Boende | 1 | 0.1 | | | |

**Table 3. Seroprevalence for different (combinations of) antibodies against Ebola virus antigens as measured by the Luminex or FANG ELISA in Health care providers from Boende, DRC.**

| | Antigen | *Cutoff | Positives n (N) | Seroprevalence % (95% conf. Int.) | Age /year | M *vs* F | Direct Contact with patients: Direct vs indirect | Working Hospital vs elsewhere | Experienced Ebola outbreak/patients vs others |
|---|---|---|---|---|---|---|---|---|---|
| | | | | | p-value | p-value | (p-value) | (p-value) | p-value |
| FANG ELISA | GP-EBOV-k | 526 EU/ml | 49 (693) | 7.0 (6.5, 10.9) | 0.89 | 0.44 | 0.52 | 0.93 | 0.94 |
| Luminex | GP-EBOV-m | 669 MFI/ 100 beads | 60 (698) | 8.6 (6.5,10.7) | **0.03** | 0.93 | 0.05 | 0.63 | 0.09 |
| | GP-EBOV-kis | 670 MFI/ 100 beads | 66 (698) | 9.4 (7.5,11.8) | 0.05 | 0.99 | 0.05 | 0.52 | **0.005** |
| | VP40-EBOV-m | 441 MFI/ 100 beads | 87 (698) | 12.4 (10.3,14.9) | 0.11 | 0.07 | 0.31 | 0.74 | 0.46 |
| | NP-EBOV-m | 602 MFI/ 100 beads | 9 (698) | 1.3 (0.6,2.6) | 0.41 | 0.40 | 0.13 | 0.26 | 0.07 |
| | GP-EBOV-m +NP-EBOV-m | C1 | 0 (698) | 0 | | | | | |
| | GP-EBOV-m +VP40-EBOV-m | C2 | 10 (698) | 1.4 (0.7,2.6) | 0.75 | 0.02 | 0.39 | 0.68 | 0.47 |
| | NP-EBOV-m +VP40-EBOV-m | C3 | 2 (698) | 0.3 (0.0,1.0) | | | | | |
| Luminex and FANG ELISA | GP-EBOV-m + GP-EBOV-k | C4 | 6 (693) | 0.8 (0.1,1.5) | | | | | |

C1 = 669MFI/100 beads for GP-EBOV-m and 602 MFI/100 beads for NP-EBOV-m.

C2 = 669MFI/100 beads for GP-EBOV-m and 441 MFI/100beads fo VP40-EBOV-m.

9.4% (66/698;95%CI 7.5–11.8) for GP- EBOV-kis, 9.4% (87/698;95%CI10.3–14.9) for VP40-E-BOV-m, and 1.3% (9/698;95%CI0.6–2.6) for NP-EBOV-m (Table 3). The seroreactivity to at least two EBOV antigens using Luminex was encountered in 1.4% (10/698;95%CI0.7–2.6) and 0.3% (2/698;95%CI0.0–1.0) of sera for VP40-EBOV-m + GP-EBOV-m and VP40-EBOV-m + NP-EBOV-m respectively. No sera tested positive for NP-EBOV-m+GP-EBOV-m.

GP-EBOV-k seroreactivity on the FANG ELISA was found in 7% (49/693; 95%CI6.5–10.9) of participants' sera. Looking at participants whose GP-EBOV seroreactivity was identified in both Luminex and FANG ELISA, 0.8% (6/693; 95%CI0.1–1.5) of the tested samples were positive by a combination of the Luminex and FANG ELISA assays. We performed seroreactivity analyses using cutoffs determined in the literature and found similar results as depicted in the (S4 Table).

A weak correlation between the FANG ELISA and Luminex was shown (k = 0.2) (Fig 1).

In seeking which participant characteristics influenced seropositivity, we observed significant differences in the seropositivity rate between HCPs and frontliners who previously made direct contact with an Ebola patient or experienced an outbreak in their hometown. When looking at the GP-EBOV-k antigen, HCPs and frontliners who previously became into contact with Ebola were significantly less likely to be seropositive compared to HCPs and frontliners who never experienced an Ebola outbreak (estimate = -1.22, std. error = 0.52, P = 0.02). When looking at the GP-EBOV-m, seropositivity status significantly decreased with age (estimate = -0.02, est.error = 0.01, P = 0.04).

## Discussion

We report the baseline seroreactivity to EBOV-m antigens in apparently healthy HCPs and frontliners enrolled in the EBL2007 vaccine trial.

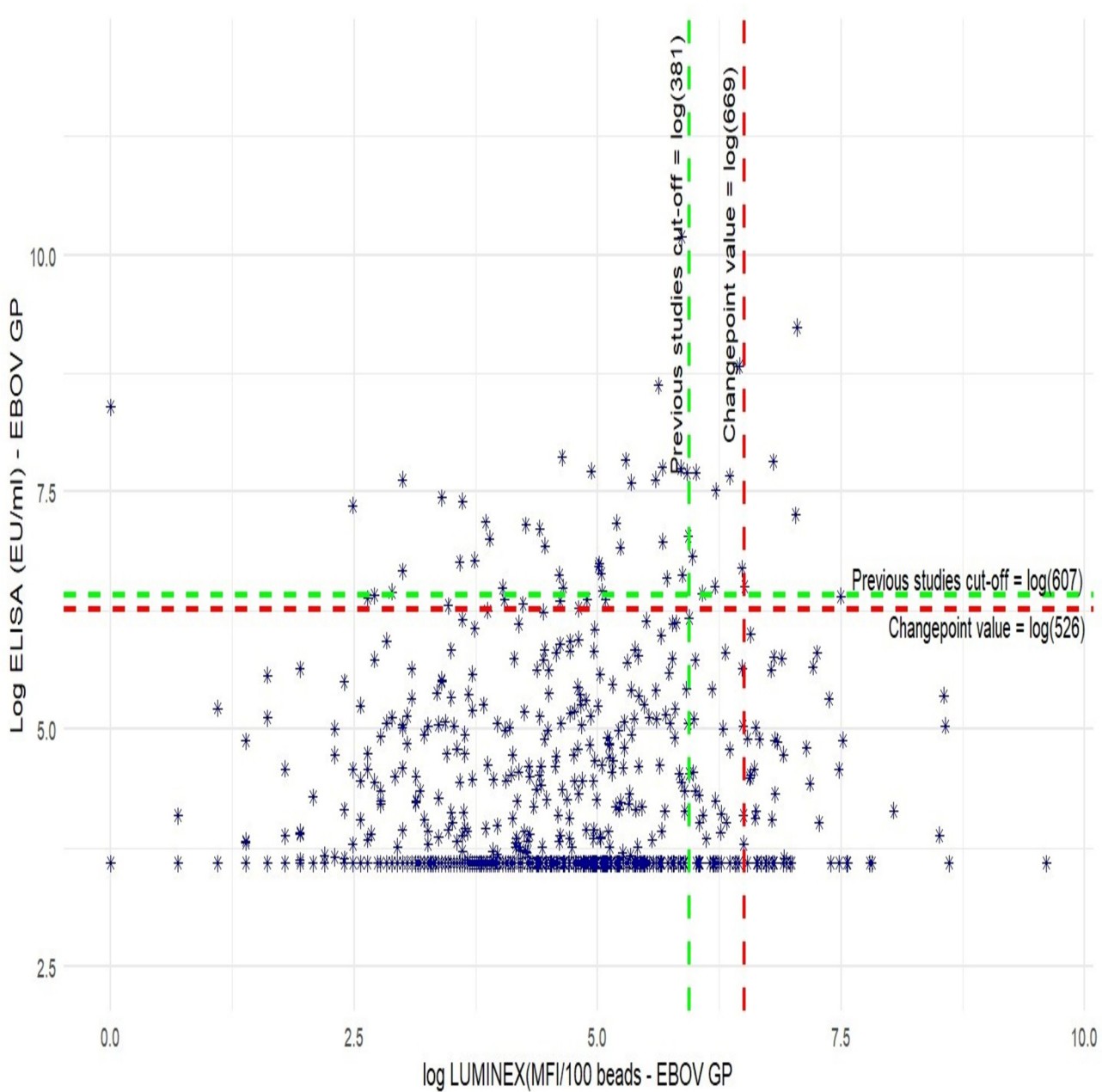

**Fig 1. Pearson correlation between FANG ELISA and Luminex.** Seroreactivity against the glycoprotein (GP) of Ebola virus in health care providers and frontliners from Boende. (A) The X-axis reports the log values of the antibody titers (IgG) as measured by Luminex in MFI/100 beads. (B) Y-axis represents antibody titers as measured by FANG ELISA in EU/ml. (C) The vertical dashed line in red represents the cutoff of the changepoint analysis and the dashed horizontal line in green represents the cutoff obtained from previous studies.

Based on seroreactivity in two different assay formats (FANG ELISA and Luminex), only a minority (0.8%) of HCPs and frontliners blood samples seroreacted to the GP-EBOV-m and GP-EBOV-k surface antigen. Similarly, a minority of participants sera tested positive to at least two antigens on the Luminex (0.3% for NP+VP40 EBOV-m and 1.4 for GP-EBOV-m+-VP40-EBOV-m). None of the participants sera tested positive to GP-EBOV-m+NP-EBOV-m.

Additionally, when we investigated whether seropositivity correlated with participants' prior exposure to EBOV-m, we did not observe a relevant positive correlation. This suggests that the majority of seropositive participants implied based on the single antigen using FANG or Luminex assays analysis are in fact false positives.

Unexpectedly, HCPs and frontliners participants who made previous contact with an Ebola case were less likely to be EBOV-seropositive than those who never became into contact. This result also suggests that the FANG ELISA is less suitable for seroepidemiological studies in African populations. Indeed, while the detection limit of the FANG ELISA (36–11 EU/mL) was established based on non-African samples, the limit needs to be increased in the African population. In this context, the Luminex multiplex assay might be much more suitable due to the use of multiple antigens that increase the specificity [40].

Overall, these results suggest that the baseline seroprevalence against EBOV-m in HCPs and frontliners in Boende is very low. Our seroprevalence estimates are much lower compared to previous serosurveys conducted after the EVD outbreak of 2014 in HCP of Boende Health District (22.5% and 28%) (GP-EBOV-m seroreactivity using ELISA) [12,27]. Our estimates are also lower than the one previously reported in Boende Health District among the general population (7%) (GP-EBOV-m seroreactivity using ELISA) a year after the 2014 Ebola epidemic [22]. The seroprevalence based on one EBOV-m antigen (GP-EBOV-m) found in this study using either ELISA or Luminex is lower than other previously reported in the Watsa Pygmy population of DRC in 2002 (18.7%) (GP-EBOV-m seroreactivity using ELISA) and the Sankuru rural population in 2007 (11%) (GP-EBOV-m seroreactivity using ELISA) [8,21].

By employing this approach, our seroprevalence estimates became comparable to those of previous serological surveys conducted in Kikwit (2.2%) (GP-EBOV-m seroreactivity by ELISA) and Kinshasa (2%) (GP-EBOV-m seroreactivity by luciferase immunoprecipitation system + neutralization) [18,41].

It is worth noting that LUMINEX built on an approach of simultaneously targeting multiple EBOV antigens, demonstrated a specificity (99.1%) and a sensitivity (95.7%) similar to higher than, respectively, the specificity (100%) and sensitivity (92.5%) of the commercial ELISA in a study [32]. The FANG ELISA was developed and validated to quantify Filovirus anti-GP-E-BOV immunoglobulin G (IgG) binding antibodies in human and non-human primate serum sample to enable bridging of immunogenicity data between humans and animal models in vaccine trials [37].

The higher seroprevalences found in other serosurveys conducted in Boende or elsewhere in the DRC may be explained by the fact that different assays were used in the different studies, other cutoff algorithms were used, and the definition of reactivity discrimination (one or two EBOV antigens) may have decreased the specificity of these assays. This could have led to overestimation of EBOV antigen seropositivity. On the other hand, it cannot be ruled out that more people were indeed infected during the 2014 outbreak in Boende and that antibody titers waned over time or at least dropped below the detection threshold, explaining the low seroprevalence that we observed. However, a number of other studies have shown IgG positivity typically prolonged to more than 10 years after an EBOV declared epidemic in an area [42,43]. It is unclear if the high compliance with infection prevention and control measures may have led to the low seroprevalence of the majority of HCPs and frontliners participants in the EBL2007 vaccine trial and serosurvey during the 2014 epidemic. This would have kept them free of EBOV exposure and might explain the low seroprevalence.

Likewise, this low prevalence may reflect a rare incidence of asymptomatic EBOV infection among HCP and frontliner population from the Boende Health District. The previous scenario may reflect a susceptibility to future outbreaks of EBOV. Yet, negative antibody titers do not rule out other types of immunity, such as T-cell immunity [44].

FANG ELISA or Luminex are assays that can only detect binding antibodies and are unable to differentiate them from neutralizing antibodies [37]. The latter are typically detected using neutralization assays, which are still considered the gold standard for serological testing [45,46]. However, such testing involves infectious cells, are labour intensive and time consuming [47]. For viruses such as EBOV, all experiments should be performed under a biosafety laboratory (BSL)-4 conditions, which are limited in availability and expensive to operate [48]. Thus, it is beneficial to use alternative neutralization assays that do not require viruses or live cells, and that can be performed in BSL-2 laboratories to assess neutralizing antibody capacity [47,49]. These alternative assays should conclude if a person with high binding antibodies against EBOV (based on FANG ELISA or Luminex) was indeed infected with the virus (although some level of cross-reactivity can never be ruled out) [29]. The challenge of comparing different serosurveys that have assessed the EBOV seroprevalence makes the implementation of international standardization of units for EBOV antibody detection and quantification of paramount importance [50].

The poor linear relationship between the two assays used (FANG ELISA and Luminex) in this serosurvey confirms that both assays likely contain many false positive results, when using single antigens. Hence, the reported seropositivity could be an effect of other filoviruses or infectious microbes, which may cause cross-reactions [51].

Limitations of our study are the lack of positive and negative control samples to determine the positive cutoff and relative long timing since the outbreak (6 years). However, in the absence of a standard serological assay for EBOV seroreactivity, Luminex can still be employed in serosurveys due to its ability to detect seroreactivity to combinations of different EBOV antigens [32].

The strength of this survey resides in the use of high cutoffs to determine the EBOV seropositivity that aligns with recommendations in EBOV serosurveys generally applied to Congolese cohorts [18,28,52]. Thus, the combination of FANG ELISA and Luminex results can be considered a starting point, showing how previous serological surveys may have overestimated the seroprevalence of EBOV in a non-exempt area. Like the recent index case of the fourteenth outbreak in DRC (Mbandaka, 2022), whose symptoms began three weeks after returning from Boende with no notion of contact with an Ebola survivor [5]. The next step could be the use of a neutralization assay for assessment of neutralizing antibodies among this population of HCP and frontliners participants in the EBL2007 trial, to further document whether or not this population of HCP is naive to EBOV exposure. Alternatively, an assessment of EBOV seroprevalence in a different population cohort closer to the time of an EVD outbreak, using negative controls, may provide insight into the utility of using Luminex or other multiple assays as the gold standard in EBOV seropositivity investigations.

The low baseline seroreactivity to EBOV antigens observed in HCP and frontliner population of Boende suggests that the majority of this population never came into contact with the virus, despite the fact the many HCP and frontliners worked during previous EBV outbreak in 2014. In the event of a future epidemic, mathematical models suggested that the vaccination rate of HCP in an infected area should be at least (30%) to prevent a major epidemic [53]. Therefore it is clear that HCP in endemic regions should be primary targets for vaccination in the frame of the Ebola epidemic preparedness plan in DRC [53].

## Conclusion

In contrast to previous studies that observed high seroreactivity against EBOV-m in Boende, our results show that the baseline seroprevalence of HCP and frontliners that reported no previous EBOV infections is low. This suggests that asymptomatic infections are unlikely to occur

or that antibodies rapidly wane after infection (or at least drop below the cutoff of detection). Irrespective of the cause, it means that the majority of HCPs in the area are likely susceptible to EVD despite the history of outbreaks in and the area of Boende. Given the high variance between seroprevalence estimates by different studies in the same region, we highlight the need for more uniform antibody assays. Neutralizing antibody quantification methods, which are inexpensive in terms of resources, are likely to be crucial for improving EVD surveillance in this region, given the high background of concomitant parasitic disease burden that can be expected to be found in the serum of this population. Low resources affordable approaches to quantifying neutralizing antibodies are likely to be crucial in enhancing surveillance of EVD disease in this region.

## Supporting information

**S1 Table. This is the baseline demographic data of participants included in the EBL2007 vaccine trial.**
(XLSX)

**S2 Table. This table presents the amount of GP-EBOV-k antibody titers measured using FANG ELISA assay.**
(XLSX)

**S3 Table. This table presents the amount of GP-EBOV-m, GP-EBOV-kis, NP-EBOV-m, and VP-EBOV-m antibody titers measured using LUMINEX assay.**
(XLSX)

**S4 Table. This table presents the EBOV seroprevalence with cut off obtained from literature for different (combinations of) antibodies against Ebola virus antigens as measured by the Luminex or FANG ELISA in Health care providers from Boende, DRC.**
(DOCX)

## Acknowledgments

The authors gratefully acknowledge the hard work and dedication of the local trial staff. The supportive role of ACE Research, DFNet Research, $Q^2$ Solutions and Institut National de Recherche Biomédicale (INRB) and all partners within the EBOVAC3 Consortium is highly appreciated.

## Author Contributions

**Conceptualization:** Trésor Zola Matuvanga.

**Data curation:** Trésor Zola Matuvanga, Joachim Mariën, Bernard Isekah Osang'ir.

**Formal analysis:** Trésor Zola Matuvanga, Joachim Mariën, Bernard Isekah Osang'ir.

**Investigation:** Trésor Zola Matuvanga, Ynke Larivière, Solange Milolo, Rachel Meta, Emmanuel Esanga, Junior Matangila.

**Methodology:** Trésor Zola Matuvanga, Joachim Mariën, Bernard Isekah Osang'ir, Jean-Pierre Van Geertruyden.

**Project administration:** Trésor Zola Matuvanga, Ynke Larivière, Vivi Maketa.

**Resources:** Solange Milolo, Rachel Meta.

**Supervision:** Trésor Zola Matuvanga, Hypolite Muhindo-Mavoko, Pierre Van Damme, Jean-Pierre Van Geertruyden.

**Validation:** Trésor Zola Matuvanga, Pierre Van Damme, Jean-Pierre Van Geertruyden.

**Visualization:** Trésor Zola Matuvanga, Ynke Larivière, Solange Milolo, Rachel Meta, Pierre Van Damme, Jean-Pierre Van Geertruyden.

**Writing – original draft:** Trésor Zola Matuvanga.

**Writing – review & editing:** Trésor Zola Matuvanga, Joachim Mariën, Ynke Larivière, Bernard Isekah Osang'ir, Solange Milolo, Rachel Meta, Emmanuel Esanga, Vivi Maketa, Junior Matangila, Patrick Mitashi, Steve Ahuka Mundeke, Hypolite Muhindo-Mavoko, Jean-Jacques Muyembe Tamfum, Pierre Van Damme, Jean-Pierre Van Geertruyden.

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
