## [Decision Letter · Decision Letter 0]

20 Mar 2023

PONE-D-23-03053Low seroprevalence of Ebola virus in health care providers in an endemic region (Tshuapa province) of the Democratic Republic of the CongoPLOS ONE

Dear Dr. Zola Matuvanga,

Thank you for submitting your manuscript to PLOS ONE. After careful consideration, we feel that it has merit but does not fully meet PLOS ONE’s publication criteria as it currently stands. Therefore, we invite you to submit a revised version of the manuscript that addresses the points raised during the review process.

We look forward to receiving your revised manuscript.

Kind regards,

Jean-François Carod

Academic Editor

PLOS ONE

Journal Requirements:When submitting your revision, we need you to address these additional requirements. 1. Please ensure that your manuscript meets PLOS ONE's style requirements, including those for file naming. The PLOS ONE style templates can be found at https://journals.plos.org/plosone/s/file?id=wjVg/PLOSOne_formatting_sample_main_body.pdf and https://journals.plos.org/plosone/s/file?id=ba62/PLOSOne_formatting_sample_title_authors_affiliations.pdf 2. Thank you for stating the following financial disclosure: "•
Initials of the authors who received each award: P.V.D, J.P.V.G•
Grant numbers awarded to each author: No 800176 (IMI-EU)•
The full name of each funder: EBOVAC3 Project, this project has received funding from the Innovative Medicines Initiative 2.This Joint Undertaking receives support from the European Union’s Horizon 2020 research and innovation pro-gramme, European Federation of Pharmaceutical Industries and Associations (EFPIA) and the Coalition for Epidemic Preparedness Innovations (CEPI)•
URL of each funder website: https://www.ebovac.org/ebovac-3/•
Did the sponsors or funders play any role in the study design, data collection and analysis, decision to publish, or preparation of the manuscript? No, the funders had no role in study design, data collection and analysis, decision to publish, or preparation of the manuscript."  Please state what role the funders took in the study.  If the funders had no role, please state: "The funders had no role in study design, data collection and analysis, decision to publish, or preparation of the manuscript." If this statement is not correct you must amend it as needed. Please include this amended Role of Funder statement in your cover letter; we will change the online submission form on your behalf. 3. We noted in your submission details that a portion of your manuscript may have been presented or published elsewhere. [- Have the results, data, or figures in this manuscript been published elsewhere? A/ the manuscript was put online in Bioarchive (as a preprint). However, it has not been peer reviewed or accepted in another journal before - Are they under consideration for publication elsewhere? No] Please clarify whether this [conference proceeding or publication] was peer-reviewed and formally published. If this work was previously peer-reviewed and published, in the cover letter please provide the reason that this work does not constitute dual publication and should be included in the current manuscript. 4. PLOS requires an ORCID iD for the corresponding author in Editorial Manager on papers submitted after December 6th, 2016. Please ensure that you have an ORCID iD and that it is validated in Editorial Manager. To do this, go to ‘Update my Information’ (in the upper left-hand corner of the main menu), and click on the Fetch/Validate link next to the ORCID field. This will take you to the ORCID site and allow you to create a new iD or authenticate a pre-existing iD in Editorial Manager. Please see the following video for instructions on linking an ORCID iD to your Editorial Manager account: https://www.youtube.com/watch?v=_xcclfuvtxQ 5. Please include captions for your Supporting Information files at the end of your manuscript, and update any in-text citations to match accordingly. Please see our Supporting Information guidelines for more information: http://journals.plos.org/plosone/s/supporting-information. 

Additional Editor Comments:

Dear Sir,

Here are the comments from the reviewers, please change accordingly and retun with your revised version.

REVIEWER ONE COMMENTS

MINOR MODIFICATIONS

I greatly enjoyed reading this fascinating manuscript. It provides a much-needed and much-overdue comparison of a multiplex assay to a popularly-used single-antigen assay for EBOV, generated interesting findings that are relevant to important public health and Ebola outbreak-related questions, and provided thought-provoking discussion and comments. I have a few comments, mostly minor.

1. General comment – there are typos throughout the paper, (examples – line 50: “yearswith”; 53: “likley" misspelled, there are others – please review the manuscript thoroughly and correct)

2. Line 287 – “GP-EBOV-m” seroreactivity on the FANG ELISA” – isn’t this supposed to be “GP-EBOV-k” for Kikwit, as described on line 180 of methods and in Table 3?

3. Line 290 “ 0.8% of the tested samples were positive”. Positive by what assay/combination of assays?

4. Lines 303-306: you note that for the GP-EBOV-k antigen – presumably this is the one tested by only the FANG assay, correct? – that HCP’s surveyed as being in previous contact with Ebola were, contrary to expectation, *less*likely to be seropositive to those who had not experienced an EBOV outbreak. This is notable and in and of itself was not discussed in the discussion section. Could this be additional supportive evidence, in addition to the correlation study itself that you performed, that the FANG essay specifically in and of itself is simply not equipped to evaluate serostatus over time? In other words, if not only did the FANG assay not correlate in the expected manner to the environmental survey but had a significant correlation in the *opposing* direction, which was not found by the LUMINEX assay, this would seem to be significant enough evidence that the FANG is simply not a useful assay in this context. If you agree with that point, I think it would be useful to point this out more explicitly in the discussion section.

5. There is a seeming discrepancy between Table 3 (lines 267-270) and the Supplementary Table (596-598). In the former the FANG ELISA antigen is listed as “GP-EBOV-k”, and the Luminex and FANG ELISA antigen is listed as “GP-EBOV-m + GP-EBOV-k” as per the methods, but in the supplement the FANG ELISA antigen is listed as “GP-EBOV-m” and the Luminex and FANG ELISA antigen is listed as “GP-EBOV-m”. Is this because the literature only has FANG ELISA against “GP-EBOV-m” whereas your assay specifically tested it against k? If so, I would clarify that, or if it’s an error, would correct it. Also the Luminex in the supplement has “GP-EBOV-k” and in Table 3 it is “GP-EBOV-kis” – not sure if those stand for the same thing but would just confirm.

6. As per comment #5, on line 323 you write “based on seroreactivity in two difference assay formats (FANG ELISA and Luminex), only a minority (0.8%) of HCPs and frontliners blood samples seroreacted to the GP-EBOV-m surface antigen.” But I thought you didn’t test the FANG ELISA against the m sAg, only the k? Please clarify.

7. It is notable that the 0.8% seroprevalence between Luminex and FANG ELISA (for whatever antigen – as per above this is unclear to me) is the same in Table 3 and Supplementary Table. This would seem to be another argument – replicability of your own findings with those based on literature cutoffs – that strengthens your argument regarding how little overlap there is between these assays. If I interpreted this correctly and you are in agreement, I would call this out more openly, as it is further evidence in support of your claims.

8. Lines 329-331 “This suggests that the majority of seropositive participants implied on the basis of the single antigen analysis are in fact false positives” – does this hold not only for FANG but also for the single-antigen Luminex analysis? If so I would just clarify that.

9. Why did you conduct the survey -- which asks questions about the past (i.e., Boende/2014-era) contacts – 1 year into the EBL2007 vaccine trial? Another way of asking: why not just ask the survey up front, when you did the serosurvey? This would presumably limit additional recollection bias, no? Some explanation seems to be needed here.

10. Related to the above, regardless of when you were conducting the survey, would note the issue of recollection bias as part of the survey conducted a year into the vaccine trial (5-6+ years after the Boende outbreak).

11. 76.5% of participants were male – would note this in the discussion as a (probably minor, but nevertheless) limitation to generalizability. Would also touch on other types of biases that may have been present in recruiting a cohort that is already participating in a vaccine trial – things like well bias, self-selection bias.

12. Lines 359-361: “it cannot be ruled out that more people were indeed infected during the 2014 outbreak in Boende and that antibody titers waned over time or at least dropped below the detection threshold, explaining the low seroprevalence that we observed.” – This is theoretically a fair argument however several longer-term studies although with smaller n have shown typically prolonged IgG positivity (Wauquier N, Becquart P, Gasquet C, Leroy EM. Immunoglobulin G in Ebola outbreak survivors, Gabon. Emerg Infect Dis. 2009 Jul;15(7):1136-7; Thomas G. Ksiazek, Cynthia P. West, Pierre E. Rollin, Peter B. Jahrling, C. J. Peters, ELISA for the Detection of Antibodies to Ebola Viruses, The Journal of Infectious Diseases, Volume 179, Issue Supplement_1, February 1999, Pages S192–S198). While you’re correct to say it cannot be ruled out, it is not necessarily the likeliest argument either – you could clarify this if you choose

13. Lines 416-418: you say that the next step would be more uniform antibody assays, and also neutralizing antibody quantification methods. However, you omit mention of another next step, which would be to do the study again, possibly with a different cohort, but both closer to the time of the outbreak and with negative controls. If your findings here were replicated, this would provide strong rationale for the Luminex itself or other similar multiplex assays as a leading candidate to be adopted as a gold standard for serosurvey purposes.

REVIEWER 2 COMMENTS

MAJOR MODIFICATIONS

1) The seropositivity is lower as compared to the previous studies where different kits were used. Did authors run some preliminary study on reference samples to compare the sensitivity and specificity of the testing assays they used with the testing assays/kits used by other investigators?

2) If not, could authors try their archived samples with the other testing assays/kits to compare the results with those obtained by authors in this study?

3) Check spelling. For example, line 123 – “targets”

Kind regards,

Dr Jean-François Carod

jfcarod@yahoo.es

PLOS ONE

Reviewers' comments:

Reviewer's Responses to Questions

**Comments to the Author**

1. Is the manuscript technically sound, and do the data support the conclusions?

Reviewer #1: Yes

Reviewer #2: Partly

2. Has the statistical analysis been performed appropriately and rigorously? 

Reviewer #1: I Don't Know

Reviewer #2: Yes

3. Have the authors made all data underlying the findings in their manuscript fully available?

Reviewer #1: Yes

Reviewer #2: Yes

4. Is the manuscript presented in an intelligible fashion and written in standard English?

Reviewer #1: Yes

Reviewer #2: Yes

5. Review Comments to the Author

Reviewer #1: I greatly enjoyed reading this fascinating manuscript. It provides a much-needed and much-overdue comparison of a multiplex assay to a popularly-used single-antigen assay for EBOV, generated interesting findings that are relevant to important public health and Ebola outbreak-related questions, and provided thought-provoking discussion and comments. I have a few comments, mostly minor.

1. General comment – there are typos throughout the paper, (examples – line 50: “yearswith”; 53: “likley" misspelled, there are others – please review the manuscript thoroughly and correct)

2. Line 287 – “GP-EBOV-m” seroreactivity on the FANG ELISA” – isn’t this supposed to be “GP-EBOV-k” for Kikwit, as described on line 180 of methods and in Table 3?

3. Line 290 “ 0.8% of the tested samples were positive”. Positive by what assay/combination of assays?

4. Lines 303-306: you note that for the GP-EBOV-k antigen – presumably this is the one tested by only the FANG assay, correct? – that HCP’s surveyed as being in previous contact with Ebola were, contrary to expectation, *less*likely to be seropositive to those who had not experienced an EBOV outbreak. This is notable and in and of itself was not discussed in the discussion section. Could this be additional supportive evidence, in addition to the correlation study itself that you performed, that the FANG essay specifically in and of itself is simply not equipped to evaluate serostatus over time? In other words, if not only did the FANG assay not correlate in the expected manner to the environmental survey but had a significant correlation in the *opposing* direction, which was not found by the LUMINEX assay, this would seem to be significant enough evidence that the FANG is simply not a useful assay in this context. If you agree with that point, I think it would be useful to point this out more explicitly in the discussion section.

5. There is a seeming discrepancy between Table 3 (lines 267-270) and the Supplementary Table (596-598). In the former the FANG ELISA antigen is listed as “GP-EBOV-k”, and the Luminex and FANG ELISA antigen is listed as “GP-EBOV-m + GP-EBOV-k” as per the methods, but in the supplement the FANG ELISA antigen is listed as “GP-EBOV-m” and the Luminex and FANG ELISA antigen is listed as “GP-EBOV-m”. Is this because the literature only has FANG ELISA against “GP-EBOV-m” whereas your assay specifically tested it against k? If so, I would clarify that, or if it’s an error, would correct it. Also the Luminex in the supplement has “GP-EBOV-k” and in Table 3 it is “GP-EBOV-kis” – not sure if those stand for the same thing but would just confirm.

6. As per comment #5, on line 323 you write “based on seroreactivity in two difference assay formats (FANG ELISA and Luminex), only a minority (0.8%) of HCPs and frontliners blood samples seroreacted to the GP-EBOV-m surface antigen.” But I thought you didn’t test the FANG ELISA against the m sAg, only the k? Please clarify.

7. It is notable that the 0.8% seroprevalence between Luminex and FANG ELISA (for whatever antigen – as per above this is unclear to me) is the same in Table 3 and Supplementary Table. This would seem to be another argument – replicability of your own findings with those based on literature cutoffs – that strengthens your argument regarding how little overlap there is between these assays. If I interpreted this correctly and you are in agreement, I would call this out more openly, as it is further evidence in support of your claims.

8. Lines 329-331 “This suggests that the majority of seropositive participants implied on the basis of the single antigen analysis are in fact false positives” – does this hold not only for FANG but also for the single-antigen Luminex analysis? If so I would just clarify that.

9. Why did you conduct the survey -- which asks questions about the past (i.e., Boende/2014-era) contacts – 1 year into the EBL2007 vaccine trial? Another way of asking: why not just ask the survey up front, when you did the serosurvey? This would presumably limit additional recollection bias, no? Some explanation seems to be needed here.

10. Related to the above, regardless of when you were conducting the survey, would note the issue of recollection bias as part of the survey conducted a year into the vaccine trial (5-6+ years after the Boende outbreak).

11. 76.5% of participants were male – would note this in the discussion as a (probably minor, but nevertheless) limitation to generalizability. Would also touch on other types of biases that may have been present in recruiting a cohort that is already participating in a vaccine trial – things like well bias, self-selection bias.

12. Lines 359-361: “it cannot be ruled out that more people were indeed infected during the 2014 outbreak in Boende and that antibody titers waned over time or at least dropped below the detection threshold, explaining the low seroprevalence that we observed.” – This is theoretically a fair argument however several longer-term studies although with smaller n have shown typically prolonged IgG positivity (Wauquier N, Becquart P, Gasquet C, Leroy EM. Immunoglobulin G in Ebola outbreak survivors, Gabon. Emerg Infect Dis. 2009 Jul;15(7):1136-7; Thomas G. Ksiazek, Cynthia P. West, Pierre E. Rollin, Peter B. Jahrling, C. J. Peters, ELISA for the Detection of Antibodies to Ebola Viruses, The Journal of Infectious Diseases, Volume 179, Issue Supplement_1, February 1999, Pages S192–S198). While you’re correct to say it cannot be ruled out, it is not necessarily the likeliest argument either – you could clarify this if you choose

13. Lines 416-418: you say that the next step would be more uniform antibody assays, and also neutralizing antibody quantification methods. However, you omit mention of another next step, which would be to do the study again, possibly with a different cohort, but both closer to the time of the outbreak and with negative controls. If your findings here were replicated, this would provide strong rationale for the Luminex itself or other similar multiplex assays as a leading candidate to be adopted as a gold standard for serosurvey purposes

Reviewer #2: 1) The seropositivity is lower as compared to the previous studies where different kits were used. Did authors run some preliminary study on reference samples to compare the sensitivity and specificity of the testing assays they used with the testing assays/kits used by other investigators?

2) If not, could authors try their archived samples with the other testing assays/kits to compare the results with those obtained by authors in this study?

3) Check spelling. For example, line 123 – “targets”

6. PLOS authors have the option to publish the peer review history of their article (what does this mean?). If published, this will include your full peer review and any attached files.

Reviewer #1: No

Reviewer #2: No

---

## [Author Response · Author response to Decision Letter 0]

20 Apr 2023

Editor(s)' Comments to Author

MINOR MODIFICATIONS

I greatly enjoyed reading this fascinating manuscript. It provides a much-needed and much-overdue comparison of a multiplex assay to a popularly-used single-antigen assay for EBOV, generated interesting findings that are relevant to important public health and Ebola outbreak-related questions, and provided thought-provoking discussion and comments. I have a few comments, mostly minor.

R/ Thank you for the opportunity to revise our paper on ‘Low seroprevalence of Ebola virus in health care providers in an endemic region (Tshuapa province) of the Democratic Republic of the Congo.’ We appreciate your insightful comments on revising the paper.

1. General comment – there are typos throughout the paper, (examples – line 50: “yearswith”; 53: “likley" misspelled, there are others – please review the manuscript thoroughly and correct)

R/ Thank you so much for catching these typos, which have been corrected. We have gone through the entire manuscript carefully and adjusted accordingly.

2. Line 287 – “GP-EBOV-m” seroreactivity on the FANG ELISA” – isn’t this supposed to be “GP-EBOV-k” for Kikwit, as described on line 180 of methods and in Table 3?

R/ It should be ‘GP EBOV-k’. It is adjusted accordingly. Line 294

3. Line 290 “0.8% of the tested samples were positive”. Positive by what assay/combination of assays?

R/ We lengthened the sentence a bit by adding “0.8% of the tested samples were positive by a combination of the Luminex and FANG ELISA assays”. Line 297

4. Lines 303-306: you note that for the GP-EBOV-k antigen – presumably this is the one tested by only the FANG assay, correct? – that HCP’s surveyed as being in previous contact with Ebola were, contrary to expectation, *less*likely to be seropositive to those who had not experienced an EBOV outbreak. This is notable and in and of itself was not discussed in the discussion section. Could this be additional supportive evidence, in addition to the correlation study itself that you performed, that the FANG essay specifically in and of itself is simply not equipped to evaluate serostatus over time? In other words, if not only did the FANG assay not correlate in the expected manner to the environmental survey but had a significant correlation in the *opposing* direction, which was not found by the LUMINEX assay, this would seem to be significant enough evidence that the FANG is simply not a useful assay in this context. If you agree with that point, I think it would be useful to point this out more explicitly in the discussion section.

R/ Thank you for this excellent observation. We have added following sentences in the discussion highlighting that FANG ELISA as single-antigen assay for EBOV is not enough equipped to evaluate sero status over time:

“Unexpectedly, HCPs and frontliners participants who made previous contact with an Ebola case were less likely to be EBOV-seropositive than those who never became into contact. This result also suggests that the FANG ELISA is less suitable for seroepidemiological studies in African populations. Indeed, while the detection limit of the FANG ELISA (36-11 EU/mL) was established based on non-African samples, this limit needs to be increased in the African population (line 341-348).

5. There is a seeming discrepancy between Table 3 (lines 267-270) and the Supplementary Table (596-598). In the former the FANG ELISA antigen is listed as “GP-EBOV-k”, and the Luminex and FANG ELISA antigen is listed as “GP-EBOV-m + GP-EBOV-k” as per the methods, but in the supplement the FANG ELISA antigen is listed as “GP-EBOV-m” and the Luminex and FANG ELISA antigen is listed as “GP-EBOV-m”. Is this because the literature only has FANG ELISA against “GP-EBOV-m” whereas your assay specifically tested it against k? If so, I would clarify that, or if it’s an error, would correct it. Also, the Luminex in the supplement has “GP-EBOV-k” and in Table 3 it is “GP-EBOV-kis” – not sure if those stand for the same thing but would just confirm.

R/ Thank you very much for catching this confusing error that is adjusted. The discrepancy between Table 3 and the supplementary Table is lifted. 

6. As per comment #5, on line 323 you write “based on seroreactivity in two difference assay formats (FANG ELISA and Luminex), only a minority (0.8%) of HCPs and frontliners blood samples seroreacted to the GP-EBOV-m surface antigen.” But I thought you didn’t test the FANG ELISA against the m sAg, only the k? Please clarify.

R/ Adjusted accordingly. This percentage is related to both antigens GP-BOV (GP-EBOV-k +GP-EBOV-m) as tested by FANG and Luminex respectively. Line 295

7. It is notable that the 0.8% seroprevalence between Luminex and FANG ELISA (for whatever antigen – as per above this is unclear to me) is the same in Table 3 and Supplementary Table. This would seem to be another argument – replicability of your own findings with those based on literature cutoffs – that strengthens your argument regarding how little overlap there is between these assays. If I interpreted this correctly and you are in agreement, I would call this out more openly, as it is further evidence in support of your claims.

R/ We agree and have added the following sentence in the discussion to support that our findings are replicable with those of previous literature: “The 0.8% seroprevalence of GP-EBOV between Luminex and FANG ELISA, regardless of the used cutoff, seems to be an additional argument further evidence in support of our claims” (Line 341-343).

8. Lines 329-331 “This suggests that the majority of seropositive participants implied on the basis of the single antigen analysis are in fact false positives” – does this hold not only for FANG but also for the single-antigen Luminex analysis? If so, I would just clarify that.

R/ Yes, it does hold for both FANG and Luminex. It is adjusted (Line 340).

9. Why did you conduct the survey -- which asks questions about the past (i.e., Boende/2014-era) contacts – 1 year into the EBL2007 vaccine trial? Another way of asking: why not just ask the survey up front, when you did the serosurvey? This would presumably limit additional recollection bias, no? Some explanation seems to be needed here.

R/ A sentence is adjusted in the discussion to reflect this further potential recollection bias in data collection. 

Limitations of our study are the lack of positive and negative control samples to determine the positive cutoff and relative long timing since the outbreak (6 years). (Line 401-402).

10. Related to the above, regardless of when you were conducting the survey, would note the issue of recollection bias as part of the survey conducted a year into the vaccine trial (5-6+ years after the Boende outbreak).

R/ This further limitation is now reflected in the discussion section. Line 408-409.

11. 76.5% of participants were male – would note this in the discussion as a (probably minor, but nevertheless) limitation to generalizability. Would also touch on other types of biases that may have been present in recruiting a cohort that is already participating in a vaccine trial – things like well bias, self-selection bias.

R/ Given that the number of male genders represents almost 2/3 of the total number of registered health care providers and frontliners in the DRC. This notwithstanding, we therefore believe that our sample was representative of the health care workers in the DRC.

12. Lines 359-361: “it cannot be ruled out that more people were indeed infected during the 2014 outbreak in Boende and that antibody titers waned over time or at least dropped below the detection threshold, explaining the low seroprevalence that we observed.” – This is theoretically a fair argument however several longer-term studies although with smaller n have shown typically prolonged IgG positivity (Wauquier N, Becquart P, Gasquet C, Leroy EM. Immunoglobulin G in Ebola outbreak survivors, Gabon. Emerg Infect Dis. 2009 Jul;15(7):1136-7; Thomas G. Ksiazek, Cynthia P. West, Pierre E. Rollin, Peter B. Jahrling, C. J. Peters, ELISA for the Detection of Antibodies to Ebola Viruses, The Journal of Infectious Diseases, Volume 179, Issue Supplement_1, February 1999, Pages S192–S198). While you’re correct to say it cannot be ruled out, it is not necessarily the likeliest argument either – you could clarify this if you choose.

We agree and have chosen to clarify this statement (on the waning immunity as possible explanation of the low seroprevalence) by adding following sentences in the discussion to put the emphasize on another argument in favour of the low seroprevalence in our serosurveyed population:

However, a number of other studies have shown IgG positivity typically prolonged to more than 10 years after an EBOV declared epidemic in an area It is unclear if the high compliance with infection prevention and control measures in the 2014 Ebola outbreak may have led to the low seroprevalence of the majority of HCPs and frontliners participants in the EBL2007 vaccine trial and serosurvey during the 2014 epidemic (Line 373-376)

13. Lines 416-418: you say that the next step would be more uniform antibody assays, and also neutralizing antibody quantification methods. However, you omit mention of another next step, which would be to do the study again, possibly with a different cohort, but both closer to the time of the outbreak and with negative controls. If your findings here were replicated, this would provide strong rationale for the Luminex itself or other similar multiplex assays as a leading candidate to be adopted as a gold standard for serosurvey purposes.

Thank you very much this helpful observation. The following sentences are added in the discussion section in regard with the proposed next steps: “Alternatively, an assessment of EBOV seroprevalence in a different population cohort closer to the time of an EVD outbreak, using negative controls, may provide insight into the utility of using Luminex or other multiple assays as the gold standard in EBOV seropositivity investigations” (Line 417-420).

REVIEWER 2 COMMENTS

MAJOR MODIFICATIONS

1) The seropositivity is lower as compared to the previous studies where different kits were used. Did authors run some preliminary study on reference samples to compare the sensitivity and specificity of the testing assays they used with the testing assays/kits used by other investigators?

R/ Thank you very much for this excellent observation that is much appreciated. 

We did not run any preliminary study on reference samples for the following reasons:

- A previous study by Ayouba A et al. J Clin Microbiol. 2016 Dec 28;55(1):165-176, already compared the results of the LUMINEX assay to commercial ELISAs (Alpha Diagnostic, San Antonio, TX) using samples from survivors of the EBOV outbreak in Guinea (2014-2016) and negative samples from patients in France. The results of this study showed that the Luminex test had a higher specificity (95.4%: CI95% 89.6-98.0) and similar sensitivity (96.8%: CI95 91.3-98. 9) compared to those of commercial ELISAs when considering seroreactivity to single GP EBOV antigen (specificity of commercial ELISA kits was 92.6(95%CI: 86.1-96.2) and sensitivity was 96.8% (95%CI: 91.3-98.9). 

- Another study by Wei Wu et al. Virus Research 2014, 187:84-90 showed lower specificity by ELISA than by using Luminex for serological detection of antibodies specific to viruses causing hemorrhagic fevers. The specificity of the Luminex test ranged from 66 to 100.00% with a sensitivity of 90 to 98%.

- A further study by Logue et al. Journal of virological methods 2018, 255, 84-90 compared FANG ELISA to a commercial ELISA (Alpha Diagnostic International, ADI). FANG ELISA was found to be largely more accurate with less regional background than ADI ELISA (widely used in previous EBOV assays)

Furthermore, a high cutoff estimate using change point analysis (Lardeux, F et al. Memórias do Instituto Oswaldo Cruz 2016, 111: 501-504.) was employed in our study for both Luminex and FANG ELISA to yield a lower GP-EBOV-m seroprevalence value (0.8%, table 3) than in previous studies. Based on this cutoff, the GP-EBOV seroprevalence using FANG and Luminex was similar (0.8% supplement table) to the one found using the literature cut-off (determined based on control samples). Hence, we believe that our findings are likely the real exposure level of these health care providers and frontliners population participants in the EBL2007 vaccine trial who would have been either quite compliant to the promoted infection control prevention measurements during the past epidemic, or who would have had their antibodies lowered over time.

Few sentences are added in the Method section in regard with Luminex and FANG performance. (Line 161-166)

2) if not, could authors try their archived samples with the other testing assays/kits to compare the results with those obtained by authors in this study?

R/Thank you very much for the request. 

Given that the assays used in this study have already been validated in other projects, and given also that our study is part of a project involving a large consortium with a budget that is already approaching its end, it will be practically not possible to carry out these analyses, which will necessarily involve logistical and financial constraints that were not foreseen at the beginning of the project.

However, we suggested in the manuscript that this may be explored in future studies (Line 433-436).

3) Check spelling. For example, line 123 – “targets”

R/ Thank you very much for capturing this typo, which is corrected.

---

## [Editor Report · Decision Letter 1]

17 May 2023

Low seroprevalence of Ebola virus in health care providers in an endemic region (Tshuapa province) of the Democratic Republic of the Congo

PONE-D-23-03053R1

Dear Dr Matuvanga,

We’re pleased to inform you that your manuscript has been judged scientifically suitable for publication and will be formally accepted for publication once it meets all outstanding technical requirements.

Kind regards,

Jean-François Carod

Academic Editor

PLOS ONE

Additional Editor Comments (optional):

Thanks to have taken all remarks into account.

The result is now publishable.

---

## [Editor Report · Acceptance letter]

26 May 2023

PONE-D-23-03053R1 

Low seroprevalence of Ebola virus in health care providers in an endemic region (Tshuapa province) of the Democratic Republic of the Congo 

Dear Dr. Zola Matuvanga:

I'm pleased to inform you that your manuscript has been deemed suitable for publication in PLOS ONE. Congratulations! Your manuscript is now with our production department. 

Kind regards, 

on behalf of

Dr. Jean-François Carod 

Academic Editor

PLOS ONE